# The Nanostructure of Alkyl-Sulfonate Ionic Liquids: Two 1-Alkyl-3-methylimidazolium Alkyl-Sulfonate Homologous Series

**DOI:** 10.3390/molecules28052094

**Published:** 2023-02-23

**Authors:** Hugo Marques, José Nuno Canongia Lopes, Adilson Alves de Freitas, Karina Shimizu

**Affiliations:** Centro de Química Estrutural, Institute of Molecular Sciences, Instituto Superior Técnico, Universidade de Lisboa, Av. Rovisco Pais, 1049-001 Lisboa, Portugal

**Keywords:** sulfonate, odd–even effect, molecular dynamics, nanostructure, coiling

## Abstract

The functionalization of polymers with sulfonate groups has many important uses, ranging from biomedical applications to detergency properties used in oil-recovery processes. In this work, several ionic liquids (ILs) combining 1-alkyl-3-methylimidazolium cations [C*_n_*C_1_im]^+^ (4 ≤ *n* ≤ 8) with alkyl-sulfonate anions [C*_m_*SO_3_]^−^ (4 ≤ *m* ≤ 8) have been studied using molecular dynamics simulations, totalizing nine ionic liquids belonging to two homologous series. The radial distribution functions, structure factors, aggregation analyses, and spatial distribution functions reveal that the increase in aliphatic chain length induces no significant change in the structure of the polar network of the ILs. However, for imidazolium cations and sulfonate anions with shorter alkyl chains, the nonpolar organization is conditioned by the forces acting on the polar domains, namely, electrostatic interactions and hydrogen bonding.

## 1. Introduction

Ionic liquids (ILs) are a class of salts characterized by their low melting point temperature [1], negligible vapor pressure [1,2,3,4,5,6], high solvation capability [1,2,3,4], high chemical [7] and thermal stability [1,3,4], and low flammability [1,3,4,6]. These properties have drawn much attention in recent years because they encapsulate some of the defining characteristics of green solvents [1,2,3,4,5,8]. Moreover, the possibility of combining different cations and anions to obtain a specific IL with a given set of attributes earned them epithets such as “designer solvents” or “tuneable materials” [3,4,5,6,7].

The most common anion families of ILs are halides (such as bromide and chloride), tetrafluoroborate, hexafluorophosphate, bis(trifluoromethylsulfonyl)imide, trifluoromethanesulfonate, and others, such as cyanamide- or phosphate-based anions [9]. Despite the wide variety of different anions that can be chosen to produce an IL, few of them include sulfonate groups, SO_3_^−^ directly attached to hydrocarbon moieties. One exception are sulfonate groups covalently bonded to cationic imidazolium rings forming zwitterionic structures with chemical properties resembling that of ILs [10,11,12,13,14].

On the other hand, the sulfonate group is often used in polymers and aromatic compounds with many advantages and applications [15,16,17,18,19,20,21,22,23,24,25,26,27,28]. In vitro studies revealed that the introduction of sulfonated groups enhances adhesion, spreading, proliferation, and osteo-differentiation in living cells [15]. Similarly, sulfonate-bearing components have anti-inflammatory properties, antibacterial activity, and higher adsorption rates of trypsin [15]. Poly(sulfobetaines) present special properties, such as antifouling, antimicrobial, strong hydration properties, and good biocompatibility, leading to their use in nanotechnology, biological and medical areas, water remediation, hydrometallurgy, and the oil industry [16]. Additionally, the same compounds are used in implant coatings, wound healing, and drug delivery systems [16]. Sulfonate-modified compounds are also effective to control regioselectivity and enhance the strength of cellular bonding [18]. Moreover, they often act as surfactants and gels [27,28]. In addition, benzene sulfonate ions mixed in solutions present good detergency, surface tension, and stability [21,23,24,26].

There are some molecular dynamics (MD) studies that address the use of sulfonate-based ILs, including the characterization of iron nanoclusters in ammonium methanesulfonate ILs [29], the organization and interactions of sulfonate-based ILs at metal surfaces [30], and the combination of ammonium and imidazolium cations with sulfonate-based anions with short alkyl side chains [31,32]. In this context, ILs containing sulfonate groups with a long alkyl moiety (ILs with large nonpolar domains) is still a topic to be further explored. Moreover, most studies analysing the influence of the alkyl chain length on the structural properties of ILs are generally focused on the cation alkyl side chains, namely, imidazolium-based ions [3,4,6,33,34,35,36,37].

This work presents a molecular dynamics simulations study of the structural properties of 1-methyl-3-alkyl imidazolium cations combined with alkyl-sulfonate anions with various alkyl side chain lengths. Two homologous series of ionic liquids were selected for this purpose, namely, 1-alkyl-3-methylimidazolium octyl sulfonates [C*_n_*C_1_im][C_8_SO_3_] and 1-octyl-3-methylimidazolium alkyl sulfonates [C_8_C_1_im][C*_m_*SO_3_] (Figure 1). These two homologous series were chosen because most previous works on SO_3_-based ILs only dealt with short alkyl chains. This contribution aims to act as a starting block for the systematic development of alkyl-sulfonate-based ILs, thus expanding the structural knowledge that already exists for other IL families.

## 2. Results and Discussion

### 2.1. Structural Characterization of [C_n_C_1_im][C_8_SO_3_] and [C_8_C_1_im][C_m_SO_3_]

The MD results were validated by comparing the simulated densities at 425 K with the corresponding values calculated by a group contribution method specific for ILs [38], since no experimental values were found for both IL series. The values determined from MD simulations and by the group contribution method are listed in Table 1 with the respective deviations. Despite the attained results being systematically higher than the expected from the group contribution method, all deviations are within the expected accuracy range for this type of volumetric estimate and end up confirming the adequacy of the simulation procedure. In both IL families, the increase in alkyl chain length causes a decrease in density. It must be stressed that [C*_n_*C_1_im][C_8_SO_3_] and [C_8_C_1_im][C*_m_*SO_3_] systems with similar overall chain lengths (equal values of *m* and *n*) have equal densities calculated from the group contribution method and also very similar densities estimated by MD. The decrease in density as the alkyl chains grow is to be expected taking into account the lower density of the alkyl moieties relative to the imidazolium ring and the sulfonate head group.

Simulation snapshots at *T* = 425 K for the nine studied [C*_n_*C_1_im][C_8_SO_3_] and [C_8_C_1_im][C*_m_*SO_3_] systems are shown in Figure 1 and highlight similar structural features. All panels of Figure 1 exhibit the characteristic nano-segregation between a polar network (red/blue) and a nonpolar domain (light/dark grey). On the other hand, there is no segregation within the nonpolar domains between alkyl chains of [C*_n_*C_1_im]^+^ and [C_8_SO_3_]^−^, nor between [C_8_C_1_im]^+^ and [C*_m_*SO_3_]^−^, which results in the formation of a single nonpolar domain composed of the aliphatic groups of both ions. As the homologous series advances, the overall size of the nonpolar domains gradually increases with extensive mixing of the alkyl side chains of both ions. Finally, if one considers the evolution of the polar network evolution along each IL series, there is no disruption of the cation–anion network as the nonpolar domain increases.

### 2.2. Total Structure Factors

The total structure factor functions, *S*(*q*), calculated from the MD trajectories are shown in Figure 2. In Appendix A, both panels of Figure 2 are combined to yield a better comparison between the *S*(*q*) functions of both series. To study the structural characteristics of the ILs at the intermolecular level, we focused only on the *q*-values in the 2.0 ≤ *q* ≤ 20 nm^−1^ region, corresponding to interionic features in real space ranging from 0.3 to 3.1 nm (*d* = 2π/*q*). All calculated *S*(*q*) functions exhibit in this range three characteristic peaks that can also be seen in other ILs with large hydrocarbon side chains (generally hexyl or larger) and that are linked to distinctive structural features. The sharp diffraction peak at the small *q*-values region (2.0 ≤ *q* ≤ 6.0 nm^−1^) is the polar–nonpolar peak (PNPP), related to the separation of different polar network strands caused by the nonpolar domains and associated with a structural heterogeneity –polar–nonpolar bicontinuous segregation. The peak at intermediate *q*-values (6.0 ≤ *q* ≤ 11 nm^−1^) is the charge-ordering peak (COP), which is a hallmark of molten salts and ILs, and is a consequence of cation–anion charge alternation within a given polar network strand. In terms of real space, this corresponds to the distance between ions with the same charge which are separated by a common counterion. The diffraction peak in the 11 ≤ *q* ≤ 20 nm^−1^ range is the contact peak (CP) and accounts for many different correlations between atoms at direct contact distances. It is a feature also present in most molecular solvents.

When the amplitudes of the PNPP, the COP, and the CP are considered along each series, it is evident that the alkyl side chain length of the ILs has a pronounced effect on the PNPP. This is easily justifiable since the PNPP is related to the morphology of the nonpolar domains (defined by the alkyl moieties of the cation and anion), whereas the COP and the CP are associated with the nature of the polar network (defined by the polar moieties of the cation and anion) and neighbouring contacts within all atoms of the IL. Figure 2 also shows that the PNPPs are present even in the ILs with the smallest alkyl-substituted cation or anion [C_4_C_1_im][C_8_SO_3_] and [C_8_C_1_im][C_4_SO_3_]. In both cases, the hydrocarbon tails from the cation and the anion can form together nonpolar aggregates that are large enough to induce polar–nonpolar nano-segregation of the bicontinuous type. Along each series, there is a monotonic increase in the PNPP intensity and an associated shift to shorter *q*-values.

In the case of the [C*_n_*C_1_im][C_8_SO_3_] series, the *q*-values are 2.9 and 2.6 nm^−1^ for [C_4_C_1_im][C_8_SO_3_] and [C_8_C_1_im][C_8_SO_3_], respectively. This represents in direct space an increase in the characteristic distance between the strands of the polar network caused by the swelling of the nonpolar domains. Such increase can be estimated as ca. (2π/2.6–2π/2.9) = 0.25 nm (or 62 pm per CH_2_ unit added along the C_4_ to C_8_ series). Taking the value of 126 pm as the increase in length of an all-*trans* alkane per added CH_2_ group [38], it can be assumed that the increase in the length of the alkyl side chain in the cation does not fully impact the increase in distance between the polar network strands separated by the nonpolar domains. The better way to accommodate this fact is to assume that there is some degree of interdigitation and/or coiling of the alkyl chains [39].

The analysis of the PNPP can be further extended if one considers systems [34] composed of the same functionalized imidazolium cations coupled with the bis(trifluoromethylsulfonyl)imide anion, [NTf_2_]^−^. Along the [C*_n_*C_1_im][NTf_2_] series, no PNPP peak is present for ILs with alkyl side chains shorter than C_5_, there is a diffraction shoulder for [C_5_C_1_im][NTf_2_], an incipient PNPP for [C_6_C_1_im][NTf_2_], and well-defined PNPPs for ILs with alkyl side chains greater than C_6_, which increase in intensity and shift to lower *q*-values from C_6_ to C_10_. The increase in the characteristic distance corresponding to the PNPP can be estimated in this case as 90 pm per added CH_2_ group, a value that is lower than the 126 pm value of the increment in an all-*trans* alkyl chain but larger than the 65 pm increment obtained in the [C*_n_*C_1_im][C_8_SO_3_] series. In the [C*_n_*C_1_im][NTf_2_] case, all chains belong to the cation and have the same length, which means that the impact of increasing the alkyl side chain is more pronounced than in the case of the [C*_n_*C_1_im][C_8_SO_3_], where such effect is “diluted” by the presence of the octyl side chains of the anion.

One can also complement these findings with comparisons with other alkylmethylimidazolium alkyl-sulfonate ILs with shorter alkyl side chains. For [C_6_C_1_im][C_4_SO_3_] [40], there is a small PNPP at 3.3 nm^−1^, while there are no PNPPs in [C_2_C_1_im][C_1_SO_3_] or [C_2_C_1_im][C_4_SO_3_] ILs [37]. This is consistent with the PNPP results obtained for the [C*_n_*C_1_im][NTf_2_] series. On the other hand, if one compares the *q*-values of the PNPPs of [C_6_C_1_im][C_4_SO_3_] and [C_8_C_1_im][C_4_SO_3_], one also obtains an increase of 65 pm per CH_2_ group for the characteristic distance corresponding to the PNPP. Similar conclusions can be drawn with respect to the effect of hydrocarbon tail in the [C_8_C_1_im][C*_m_*SO_3_] series.

Furthermore, imidazolium-based ionic liquids with relatively long alkyl chains combined with anions such as halides, [BF_4_]^−^ or [PF_6_]^−^ and [NTf_2_]^−^ tend to show common features in the lower *q*-values of the structure factor intensity [41,42,43]. The mesoscopic domain in ionic liquids can by no means be considered spatially homogeneous and is recognized to retain a high degree of hierarchical organization [44,45].

In both [C*_n_*C_1_im][C_8_SO_3_] and [C_8_C_1_im][C*_m_*SO_3_] series, the observed PNPPs indicate the existence of bicontinuous mesoscopic subphases in all studied ILs. There is however a difference between the two series: Although the increase in the PNPP intensity is quite gradual along the [C*_n_*C_1_im][C_8_SO_3_] series (Figure 2a), the intensity of the PNPP along the [C_8_C_1_im][C*_m_*SO_3_] series (Figure 2b) is monotonous but is less pronounced between C_4_-C_5_ and C_6_-C_7_ and more pronounced between C_5_-C_6_ and C_7_-C_8_. This odd–even chain length effect has already been observed for different IL homologous series in different thermophysical properties, such as molar volume, salting-out ability, degree of ionization, molar conductivity, and diffusion coefficients [46]. The underlying causes of this odd–even effect that impacts the molecular packing and structure of ILs are a combination of inter- and intramolecular interactions [47,48,49].

Regarding the charge-ordering peak (COP), the maxima was obtained through deconvolution of the shoulder in Figure 2, and the *q*-value was found to be ca. 9.30 nm^−1^ for all ILs. This *q*-value corresponds to a real-space distance of ca. 0.68 nm (*d* = 2 π/*q*) between ions of the same sign separated by a bridging counterion, with the aim of achieving local electroneutrality conditions. As stated before, COP is a distinctive feature of the three-dimensional arrangement of the polar network of ILs and molten salts and is absent in molecular solvents. Shifts in the COP are usually correlated to changes in the characteristic distances between the ions of the same charge within the polar network. There is no significant displacement in *q*-values for both IL series, indicating that the arrangement between imidazolium rings and SO_3_^−^ anions resulting from electrostatic interactions and hydrogen bonding are not affected by changes in the nonpolar subphase. In principle, it should be expected slightly larger polar characteristic distances with the increase in the tail length. However, the nonpolar moiety of the counterion has already a significative volume in both IL series. This fact makes the increase in the nonpolar part of the corresponding ion less substantial, because there is not much more reorganization to be made near the polar network in order to accommodate the progressively larger nonpolar domains. With respect to the COP amplitude, the trend is the opposite of that observed for the PNPP. The PNPP describes the scattering caused by the nonpolar domains, whereas the COP accounts for the scattering along the polar network of the IL, so it is expected that the COP intensity decreases as the weight of nonpolar contributions increases, as seen for both the alkyl chain length dependency of the cation and the anion. The same trend was observed for the 1-alkyl-3-methylimidazolium bis(trifluoromethylsulfonyl)imide IL homologous series [34,50].

Finally, the contact peak (CP) is related in real space to distances between adjacent atoms belonging to different ions or different moieties within the same ion. In both series, there is a displacement to shorter *q*-values (ca. 0.2 nm^−1^), negligible when compared to the change in the PNPP. It is hard to assign which factors contribute to this shift, given the plethora of pair correlations that contribute to the CP. Nevertheless, the shift of the CP scattering to shorter reciprocal distances in function of the alkyl chain length reflects a less efficient packing and is corroborated by an equivalent decrease in density, since the longer alkyl tails have more freedom to move and accommodate within the nonpolar subphase. Further discussion about the contributions to the COP and the CP is given below, when radial pair distribution functions for the different ILs are considered.

The comparison of the PNPP for IL analogues in both families (cation C_4_–anion C_4_, cation C_5_–anion C_5_, etc.) revealed that its maximum is at slightly larger *q*-values for the [C_8_C_1_im][C*_m_*SO_3_] family members. Here it is noticeable that the morphology of the building blocks of the polar part can play a role in the organization of the nonpolar domain. The lower *q*-values of the PNPP maxima for [C*_n_*C_1_im][C_8_SO_3_] ILs, associated with larger nonpolar subphases in direct space, indicate that bulky polar heads such as cation imidazolium rings are more difficult to accommodate within the polar network than the SO_3_^−^ groups, thus restricting the organization of the short hydrocarbon tails to some preferential locations at the vicinity of the polar domains. As the chain length increases and becomes more flexible, this restrictive effect is attenuated, and for C*_n_* ≥ 6 the difference in *q*-values for PNPP between the families no longer exists. Regarding the COP and CP intensities (Appendix A), there are no sizable differences between the IL analogues, since strong segregation between polar and nonpolar domains occurs for both cases, with virtually no perturbation of the polar network as the nonpolar domain increases.

### 2.3. Pair Distribution Functions

The presence of a continuous polar network can be analysed for all studied ILs in real space using the corresponding cation–anion, anion–anion, and cation–cation radial pair distribution functions (RDFs). Figure 3 depicts the *g*(*r*)s for selected interaction centres of the polar part, namely, the centre of mass of the imidazolium ring and the sulphur atom of the SO_3_^−^ group. One distinct structural feature seen in the *g*(*r*) of ILs or molten salts is their opposition-of-phase character. This “fingerprint” is defined by a charge-ordering pattern that emerges on ILs to keep local electroneutrality conditions [51]. The nature of the Coulomb interactions dictates that the largest peak of the three *g*(*r*) functions shown in each graph of Figure 3 corresponds to cation–anion interactions. As stated before, the COP *q*-value can be related to the direct-space distance of the characteristic wavelength of the polar network. For all studied ILs, this value is ca. 0.68 nm (corresponding to a *q*-value of 9.30 nm^−1^) and is depicted in Figure 3 as vertical lines delimiting the periodic wavelengths of the polar network. These gridlines do not substantially differ from one another. Apart from the increasing intensity of the cation–anion peak with the alkyl chain length, the RDFs of the polar part do not exhibit substantial differences, as already indicated by the behaviour of the COP band in all the analysed *S*(*q*) functions, meaning that the growth of the nonpolar domains along the homologous series has little effect on the polar network in terms of accommodating these nonpolar moieties. In other homologous series based on imidazolium cations and fluorinated anions, such as [C*_n_*C_1_im][BF_4_], [C*_n_*C_1_im][PF_6_], [C*_n_*C_1_im][CHF_2_CF_2_SO_3_], and [C*_n_*C_1_im][NTf_2_] [52,53,54], the RDFs indicated some stretching of the polar network in order to fit the growing nonpolar subphase. As discussed before, the absence of COP shifts means that since the charged parts of the cation and anion are always attached to an alkyl chain, that is, at least four carbons atoms long, there is no rearrangement of the polar network as the alkyl chains increase from C_4_ to C_8_ along the studied homologous series: No extensive reorganization of the polar network is needed to accommodate the extra alkyl moieties that just go into the middle of the nonpolar domains, far from the polar moieties of the ions.

The integration of the first peaks in *g*(*r*) of the polar part provides the number of neighbours present in the first shell of a given ion. In both IL series, the average number of neighbours between imidazolium rings and the –SO_3_^−^ groups remains around 4.3. The reasons for this invariability of the polar network can be better understood by inspecting the spatial distribution functions (SDFs). Appendix A depict the SDFs for the polar part of [C_4_C_1_im][C_8_SO_3_] IL (for the sake of simplicity, only this compound will be reported, as the remaining systems presented similar results and conclusions). The SDFs around the polar parts revealed that the SO_3_^−^ groups are located preferentially at the imidazolium ring plane near the acidic hydrogen at C-2, and less preferentially in the vicinity of the C-5 hydrogen, farther from the cation alkyl chain. These preferential positions of the anion around the cation are insensitive to the side chain length and point out a significative contribution of H bonding to the stability of the polar domains. In other words, the ordering of the polar network is primarily defined by electrostatic interactions and H bonding and is unaffected by the growth of nonpolar domains in both the [C*_n_*C_1_im][C_8_SO_3_] and [C_8_C_1_im][C*_m_*SO_3_] IL series.

As regards the nonpolar domains, and since the density of the IL series decreases with the increase in the alkyl chain length, we must expect that the nonpolar subphase is accounting for all the expansion in volume that leads to lower densities. Figure 4 shows the *g*(*r*)s between terminal C atoms of the alkyl side chains of the imidazolium cations and of the sulfonate anions. In Figure 4a, the terminal CH_3_ group of the alkyl chain in the imidazolium cations is shown in two different situations: the first is the evolution of *g*(*r*) within the [C*_n_*C_1_im][C_8_SO_3_] series, and the second is the behaviour of the [C_8_C_1_im]^+^ cation in [C_8_C_1_im][C*_m_*SO_3_] family, where the alkyl side chain length of the cation was kept constant. The first peak of the *g*(*r*) functions increases as the alkyl chain increases from C_4_ to C_8_ in the [C*_n_*C_1_im][C_8_SO_3_] series and then continues to increase as one goes from C_8_ to C_4_ in the [C_8_C_1_im][C*_m_*SO_3_] series. This peculiar behaviour can be easily explained if one considers that the probability of contact between two terminal methyl groups of the cation is lower in [C_4_C_1_im][C_8_SO_3_] than in [C_8_C_1_im][C_8_SO_3_] due to the fact that in the former IL the butyl groups are “lost” in the middle of the longer octyl groups of the anion, whereas in the latter the octyl groups of the cation are as big as those of the anion. When the [C_8_C_1_im][C*_m_*SO_3_] series is considered, the intensity of the first peak continues to rise as the alkyl side chain of the anion starts to decrease due to the fact that now the long octyl chains of the cation start to contact each other more often due to the receding length of the anion side chains. Likewise, Figure 4b depicts the same two scenarios for the alkyl-sulfonate anion with similar trends.

Figure 4c portrays the *g*(*r*) between terminal CH_3_ groups of both ions. The maxima of the first peak in all *g*(*r*)s fall at the same distances (ca. 0.42 nm), indicating no segregation between ion tails. For both families, there is a residual displacement in the first peak to larger distances going from C_4_ to C_5_ and then a continuous rise in its intensity and broadening until C_8_. Likewise, as the chain length increases, the *g*(*r*) starts exhibiting an oscillatory character at intermediate distances (1.5 nm ≤ *r* ≤ 3 nm), underlining two types of competing interactions: When the alkyl group is short [55], its organization is determined by the charged head groups of the imidazolium cation and SO_3_^−^ anion. An interesting pattern is unravelled when the interactions between terminal carbon atoms of the tails of both ions are compared simultaneously. Figure 4c shows that the *g*(*r*) functions are not similar for analogue ILs (*n* = *m*), but rather for ILs where the imidazolium tail has one more methyl group than the sulfonate tail (*n* = *m* + 1). For instance, [C_5_C_1_im][C_8_SO_3_] and [C_8_C_1_im][C_4_SO_3_] have similar *g*(*r*)s, as well as [C_6_C_1_im][C_8_SO_3_] and [C_8_C_1_im][C_5_SO_3_]. After that, the *g*(*r*) functions of [C_7_C_1_im][C_8_SO_3_], [C_8_C_1_im][C_8_SO_3_], [C_8_C_1_im][C_7_SO_3_], and [C_8_C_1_im][C_6_SO_3_] are all clumped together, most likely due to the nonpolar moieties in these ILs being voluminous and far enough from the polar network. Finally, [C_4_C_1_im][C_8_SO_3_] does not have any other IL with a similar *g*(*r*). This gap between IL analogues seen in the profiles of the pair distribution functions results from the size differences among the polar parts of [C*_n_*C_1_im]^+^ cations and [C*_m_*SO_3_]^−^ anions. In addition, the sulfonate group is more flexible than the imidazolium ring, with more freedom to adopt different conformations within the polar domains and with its methylene tails counterbalancing the short-range interactions sooner than in the cation. In other words, the flexibility and interactions along the alkyl side chains of the cation and anion are not exactly the same, and a “true” alkyl side chain (free from constraints dictated by the organization of polar moiety) starts earlier for the sulfonate chains than for their imidazolium counterparts.

As the shift of PNPP to shorter *q*-values in *S*(*q*)s suggested some interdigitation or coiling with increasing alkyl chain length, we turned our attention to the structural analysis of the alkyl chain conformation. The probability distribution functions of the intramolecular distances between ion headgroups (S atom in case of sulfonate anion and N atom attached to the alkyl side chain in imidazolium cation) and the respective terminal methyl group of the aliphatic tail are presented in Figure 5. This headgroup–CH_3_ distance can also be viewed as an effective length of the alkyl tail. The two main bands seen in the distribution functions indicate the presence of an all-*trans* population, characterized by conformers with longer headgroup–CH_3_ distances, and a gauche/eclipsed population, where the alkyl backbone has more freedom to engage in bent or coiled conformations. The d(headgroup–terminal–CH_3_) functions along the series show that there is a quite representative population of all-*trans* conformers in cations and anions with shorter tails (C_4_-C_5_). As the hydrocarbon tail length increases in both IL series, the narrow distributions are giving place to broadened bands, with a gradual replacement of the all-*trans* population by alkyl chains with more conformational disorder. It is interesting to note that the C_4_ chains in the sulfonate anions show less conformational freedom than their C_4_ counterparts in the imidazolium cations.

Figure 6 shows the fraction of all-*trans* conformers for cations and anions in [C*_n_*C_1_im][C_8_SO_3_] and [C_8_C_1_im][C*_m_*SO_3_] IL families. The fractions were determined from the probability distribution functions of the dihedral angles of the alkyl chains presented in Appendix A. Both series exhibited a similar decrease in the all-*trans* population as the alkyl side chain length of the ion increases. The inspection of the torsion angles indicated that each dihedral contributes with ca. 70% to the all-*trans* conformer. In general, the terminal dihedrals of the alkyl backbone of cations and anions have the lowest contribution to the fully extended chain. However, there are interesting differences concerning the odd-length derivatives in [C_8_C_1_im][C*_m_*SO_3_] ILs, with the series showing an odd–even effect in the conformer population. In comparison with the [C*_n_*C_1_im][C_8_SO_3_] analogues, the members of the [C_8_C_1_im][C*_m_*SO_3_] family whose sum 8 + *n* is an odd number have a slightly smaller preference for fully extended zigzag conformers in anions and cations. These observations, together with the *S*(*q*) results, point out differences in the periodic packing patterns of [C*_n_*C_1_im][C_8_SO_3_] and [C_8_C_1_im][C*_m_*SO_3_] ILs.

### 2.4. Aggregation Analysis

The probability distribution functions of nonpolar aggregate sizes are shown in Figure 7. Here, the nearest neighbour search criterium used for clustering was the arbitrary presence of alkyl chains of cations and/or anions. The distance threshold of 0.50 nm was taken from the first minimum in the CT-CT *g*(*r*)s of all ILs. The size indicates that only clusters with large number of mixed aliphatic chains of cations and anions are present in both families. Similarly, the cluster size of the nonpolar domain increases with the aliphatic chain length, in line with other alkyl-functionalized cations and anions used in ILs [56]. Since the counterion has a sizeable alkyl group in both families [34], the continuous nonpolar subphase is already present even for C_4_, the smallest member of the homologous series. Still, there are subtle differences dictated by the polar headgroups. The broader aggregation distributions for members C_4_ and C_5_ of the [C*_n_*C_1_im][C_8_SO_3_] family (Figure 7a) confirm that the morphology of the imidazolium ring combined with the electrostatic interactions plays a role in the accommodation of the shorter aliphatic chains within the nonpolar domains. On the other hand, the threshold between long-range and short-range interactions is reached sooner on [C_8_C_1_im][C*_m_*SO_3_] series (Figure 7b), with only the C_4_ member exhibiting a distribution profile with *P*(*n_a_*) < 0.5 at the maximum. The aggregation analyses of the polar domains of both IL families (not shown) indicated that all ion pairs belong to a single aggregate, meaning that the increase in alkyl chain length up to C_8_ does not induce any polar network breakdown in [C*_n_*C_1_im][C_8_SO_3_] and [C_8_C_1_im][C*_m_*SO_3_] series.

With respect to the direct contact neighbours *N*_i_ (Figure 8), the average number within the polar network is ca. 4.3 along the whole homologous series for both cation and anion alkyl side chain length dependencies. The polar network is subjected to negligible swelling or breakdown because the nonpolar moiety of the counterion is large enough, and there is no need for extensive rearrangements to accommodate the increasing nonpolar domains of the ion. Regarding the nonpolar domains, the increase in *N*_i_ reflects the increment of contact points of the corresponding ionic alkyl chain with other nearby chains within the nonpolar domains. As the aliphatic chain increases, the nonpolar moiety of the ion tends to occupy interaction sites previously in use by the nonpolar domains of the counterion. Lastly, the increase in *N*_i_ seen in mixed imidazolium–sulfonate nonpolar aggregates is a consequence of the growth of the overall volume occupied by the nonpolar moieties, which in turn yields lower *q*-values in *S*(*q*).

## 3. Computational Procedure

All MD simulation runs were carried out using the GROMACS 2020 (Uppsala University, Uppsala, Sweden) and DL_POLY 2.20 (Daresbury Laboratory, Daresbury, UK) packages [57,58]. The IL ions were modelled using the previously reported CL&P atomistic force field [59,60], specifically built to encompass the entire IL homologous series. All simulations were carried out at 425 K in periodic cubic boxes consisting of systems with 1200 ion pairs. The box length of the final simulation runs for the different systems is presented in Table 1. Cutoff distances of 1.6 nm were used for electrostatic and repulsive–dispersive interactions, with the Ewald summation technique applied to account for interactions beyond that distance (accuracy of 10^−5^). The equilibrium conditions were attained using the following steps: (i) equilibrations started from initial low-density configurations created with the Packmol version 15.287 (University of Campinas, Campinas, Brazil) and fftool 2022 software (École Normale Supérieure de Lyon, Lyon, France) [61,62], with ions placed at random in the cubic boxes under canonical ensemble; (ii) this was followed by 4 ns equilibrations through the use of temperature annealing ranging from 300 to 500 K; (iii) further equilibrations were implemented for 12 ns under isothermal–isobaric (*N*-*p*-*T*) ensemble conditions (under the action of Nosé–Hoover thermostat at 425 K and Berendsen barostat, with time constants of 0.5 ps and 4 ps, respectively); (iv) subsequent simulation runs were used to produce equilibrated systems at 425 K and Parrinello-Rahman barostat. The last production step was performed for at least 8 ns.

Pair distribution functions, *g*(*r*), were calculated for selected pairs of the different IL systems with the aim of underlining specific interactions. Structure factors, *S*(*q*), calculated from the appropriate Fourier transforms of the pair distribution functions were used to unravel the structural characteristics of each system. Aggregate size distributions and an average number of neighbours were also calculated. To further analyse the results, spatial pair distribution functions (SDFs) were attained as well [63,64]. Supplementary information about the used algorithms and procedures can be found elsewhere [34,65].

## 4. Conclusions

We performed a series of analyses for members of [C*_n_*C_1_im][C_8_SO_3_] and [C_8_C_1_im][C*_m_*SO_3_] ionic liquid families, with 4 ≤ *n, m* ≤ 8, based on molecular dynamics simulation trajectories. The total structure factors and radial distribution functions point out that the increase in alkyl chain length has very little influence on the arrangement of the anion headgroups in the vicinity of the imidazolium rings, revealing a significative contribution of the H bonding to the stability of the polar domain, while the IL density decreases. This variation observed in density can be attributed to a change in the alkyl conformer population, with a gradual replacement of the all-*trans* conformers by alkyl chains with more freedom to diffuse. The aggregation results indicate that the accommodation of the shorter aliphatic chains is affected by the morphology of the imidazolium ring and by the electrostatic interactions that play a role in the organization of the polar part. Surprisingly, the first sharp diffraction peak data in *S*(*q*)s as well as the fraction of all-*trans* conformers for cations and anions display evidences of odd–even effect in [C_8_C_1_im][C*_m_*SO_3_] IL family, resulting from differences in the periodic packing patterns. In summary, the results provide important structural information about these less explored alkyl-functionalized sulfonate-based ILs.

Overall, the [C*_n_*C_1_im][C_8_SO_3_] and [C_8_C_1_im][C*_m_*SO_3_] ionic liquid families show mutual features compared with other ionic liquids based on imidazolium cation [66,67,68]. Ionic liquids do not occur in natural environments [69] and thus can be highly persistent in nature [70,71,72]. In light of that, the alkyl-sulfonate-based ionic liquids can be seen as an eco-friendly alternative for potentially harmful ionic liquids.

## Data Availability

Not applicable.

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
