# Peer review of "The Nanostructure of Alkyl-Sulfonate Ionic Liquids: Two 1-Alkyl-3-methylimidazolium Alkyl-Sulfonate Homologous Series"

_molecules, 2023, doi:10.3390/molecules28052094_

Round 1

Reviewer 1 Report

The present manuscript "The nanostructure of alkylsulfonate ionic liquids: Two
1-alkyl-3-methylimidazolium alkylsulfonate homologous series"  by Marques at al. presents a systematic study of the nanomorphology of the two groups of the alkylsulfonate anion ILS.
The article reports the MD data and their detailed analysis. All structural functions are carefully analyzed and well represented in figures.
The development of nanosegregation in long-chain alkyl ILs is well studied both experimentally and theoretically. However  the results presented in this article contribute to the fine understanding of this latter phenomenon.
I suggest this manuscript for publication in presented form.

Author Response

We thank the reviewer for her/his very positive statements concerning the quality of our work.

Reviewer 2 Report

This is interesting research with a possible application of obtained results in medical and personal care products. However, it lacks some necessary discussions and deep insights into the possible applications of obtained results. The serious limitation of this manuscript is the sketchy description of the application of these compounds in practice.

Please explain, are the compounds covered in the manuscript both or are they synthesized in the laboratory?

Also, the authors should provide more comparisons to the previous work that is done in this specific field (for example comprehensive table).

Authors should explain why they chose these specific ILs, is this choice based on previous experience or a literature survey?

Self-citation of some coauthors is high.

Summarizing, due to lack of detailed elaboration of obtained results. I recommend the publication of this manuscript after minor revision.

Author Response

This is interesting research with a possible application of obtained results in medical and personal care products. However, it lacks some necessary discussions and deep insights into the possible applications of obtained results. The serious limitation of this manuscript is the sketchy description of the application of these compounds in practice.

We thank the reviewer for her/his very positive statements concerning the quality of our work.

Please explain, are the compounds covered in the manuscript both or are they synthesized in the laboratory?

ANSWER: We have studied these compounds only by molecular dynamics simulations. We included the following sentence in the beginning of the Results and Discussion (page 2, lines 71-73).

“The MD results were validated by comparing the simulated densities at 425 K with the corresponding values calculated by a group contribution method specific for ILs [38], since no experimental values were found for both IL series.”

Also, the authors should provide more comparisons to the previous work that is done in this specific field (for example comprehensive table).

ANSWER: We included the following sentence in the structure factor discussion (page 5, lines 170-174):

Furthermore, imidazolium-based ionic liquids with relatively long alkyl chains combined with anions such as halides, [BF4]- or [PF6]- and [NTf2]- tend to show common features in the lower q-values of the structure factor intensity [41-43]. The mesoscopic domain in ionic liquids can by no means be considered spatially homogeneous and is recognized to retain a high degree of hierarchical organization [44,45].

Authors should explain why they chose these specific ILs, is this choice based on previous experience or a literature survey?

ANSWER: There are just few studies with imidazolium cations and sulfonate-based anions with short alkyl side chains. In this context, we have chosen ILs containing sulfonate groups with a long alkyl side chain, since the MD results could provide new insights into this field. Therefore, we included the following sentence in the conclusion (page 12, lines 456-460):

Overall, the [CnC1im][C8SO3] and [C8C1im][CmSO3] ionic liquid families show mutual features compared with other ionic liquids based on imidazolium cation [66-68]. Ionic liquids do not occur in natural environments [69], and thus can be highly persistent in nature [70-72]. In light of that, the alkylsulfonate-based ionic liquids can be seen as an eco-friendly alternative for potentially harmful ionic liquids.

Self-citation of some coauthors is high.

ANSWER: We did not notice this, and we thank the referee to call our attention. In this review process, we included some more references to answer the referees's comments and we believe that the self-citations are attenuated now.

Summarizing, due to lack of detailed elaboration of obtained results. I recommend the publication of this manuscript after minor revision.